# Cardiovascular consequences of maternal obesity throughout the lifespan in first generation sheep

**Christopher L. Pankey**[1]*, **Qiurong Wang**[2], **Jessica King**[1], **Stephen P. Ford**[2]

**1** Department of Biomedical Science, West Virginia School of Osteopathic Medicine, Lewisburg, West Virginia, United States of America, **2** Department of Animal Science, University of Wyoming, Laramie, Wyoming, United States of America

* cpankey@osteo.wvsom.edu

**Data Availability Statement:** The data underlying the results presented in the study are available from the following reference: Pankey, Chris (2022) "Maternal Obese Ovine Cardiovascular Data", Mendeley Data, V1, doi: 10.17632/x8726jyg2v.1.

## Abstract

Obesity continues to be a significant global health issue and contributes to a variety of comorbidities and disease states. Importantly, obesity contributes to adverse cardiovascular health outcomes, which is the leading cause of death worldwide. Further, maternal obesity during gestation has been shown to predispose offspring to adverse phenotypic outcomes, specifically cardiovascular outcomes. Therefore, we hypothesized that diet-induced obesity during gestation would result in adverse cardiovascular phenotypes in first-generation offspring that would have functional consequences in juvenile and advanced ages. Multiparous Rambouillet/Columbia cross ewes (F0) were fed a highly palatable, pelleted diet at either 100% (CON), or 150% (OB) of National Research Council recommendations from 60 days prior to conception, until necropsy at d 135 (90%) of gestation (CON: n = 5, OB: n = 6), or through term for lambs (F1: 2.5 mo. old; CON: n = 9, OB: n = 6) and ewes (F1:9 years old; CON: n = 5, OB: n = 8). Paraffin-embedded fetal aorta section staining revealed increased collagen:elastin ratio and greater aortic wall thickness in OBF1 fetuses. Invasive auricular blood pressure recordings revealed elevated systolic blood pressure in OBF1 lambs, but no differences in diastolic pressure. In aged F1 ewes, systolic and diastolic blood pressures were reduced in OBF1 relative to CONF1. Echocardiography revealed no treatment differences in F1 lambs, but F1 ewes show tendencies for increased end systolic volume and decreased stroke volume, and markedly reduced ejection fraction. Therefore, we conclude that maternal obesity programs altered cardiovascular development that results in a hypertensive state in OBF1 lambs. Increased cardiac workload resulting from early life hypertension precedes the failure of the heart to maintain function later in life.

## Introduction

The obesity epidemic has tripled worldwide since 1975 [1]. Increased adiposity puts individuals at risk for comorbid, chronic conditions such as hypertension, type 2 diabetes, stroke and coronary artery disease [2]. The Framingham Heart Study demonstrated that individual

**Funding:** This study was supported by the Dual Purpose with Dual Benefits grant from the National Institutes of Health (https://grants.nih.gov/funding/index.htm), grant number R01 HD070096-01A1, awarded to SPF. The funders had no role in study design, data collection and analysis, decision to publish, or preparation of the manuscript.

**Competing interests:** The authors have declared that no competing interests exist.

obesity is associated with an increased risk of heart failure and not just in cases of severe obesity [3]. Proposed mechanisms behind heart failure secondary to obesity include multifactorial processes, as increased cardiac output and blood pressure have been noted in obese patients [4].

Obesity also has significant implications in reproductive settings. In 2015, the CDC reported 48% of women gained more than the recommended amount of weight during their pregnancies [5]. In 2020, an estimated 31% of women of reproductive age were obese in the United States [6]. In addition to the maternal implications, maternal obesity also impacts the fetus throughout gestational development. The developmental origins of health and disease (DOHaD) hypothesis proposes that altered developmental environments, including those associated with maternal obesity, drive adaptive developmental responses in the fetus which can be detrimental [7]. Specifically, maternal obesity during pregnancy is associated with the abnormal development of cardiometabolic organs including the heart, liver, pancreas, and others, resulting in adverse metabolic phenotypes in offspring [8, 9].

When examining the effects of maternal obesity on cardiac development of the fetus, prior studies have reported that there is an increased likelihood of early death secondary to cardiovascular complications for offspring born to mothers with an elevated BMI [10]. In rodent models, maternal obesity has been shown to have an increased risk of ventricular hypertrophy [11]. Rat models also demonstrated elevated systolic blood pressure in adult offspring of dams fed a high fat diet during gestation [12]. Additional mouse studies have suggested that offspring of obese mice have both systolic and diastolic dysfunction that eventually leads to heart failure, as the observed ventricular hypertrophy in these models is not sustainable as a long-term compensatory mechanism [13].

We have developed an ovine model of diet-induced maternal obesity in an attempt to investigate the cardiovascular consequences of maternal obesity. We propose this as a strong model for investigating our research questions given the parallels observed between sheep and humans. Specifically, ovine models have a similar temporal development, typically have single or twin pregnancies, and share similar maternal:fetal body mass ratios as humans. The model in this study has been used extensively to examine maternal obesity during gestation [14], as it produces obese ewes [15], but F1 offspring that exhibit similar weight, stature, and metabolic phenotypes until they are exposed to a metabolic stressor [16].

Prior studies from our group demonstrated that fetal ovine ventricular tissue has increased inflammatory cytokine profiles resulting from maternal overnutrition in offspring from obese ewes (OBF1) [17]. We have also reported that OBF1 ewes have myocardial inflammation and fibrosis in adulthood, which was attributed to increased glucocorticoid signaling [18]. Consequently, OBF1 lambs are predisposed to cardiovascular disease and impaired cardiac function as demonstrated by decreased contractility [19]. When exposed to high workload stress, OBF1 ewes demonstrated an increase left ventricular pressure. These OBF1 ewes were reported to have reduced activity of the AMP kinase which can potentially account for the impaired cardiac function during high workload [20].

Our model has also demonstrated adverse cardiovascular programming in OBF1 compared with CONF1 at fetal and mid-life stages [18, 21]. In those studies, OBF1 fetuses showed increased heart, left ventricle (LV), and right ventricle (RV) mass, even when corrected for fetal body mass, at mid (50%) and late (90%) gestation [17, 20, 22–24]. For the current study, we hypothesized that diet-induced obesity during gestation would result in adverse cardiovascular phenotypes in first-generation offspring that would have functional consequences in juvenile and advanced ages.

## Materials and methods

### Animals

All animal procedures were approved by the Animal Care and Use Committee at the University of Wyoming and conducted in AALAC accredited facilities. Multiparous Rambouillet/Columbia cross ewes were bred to a single ram to produce first generation (F1) fetuses and lambs. Ewes were randomly assigned to either a control (CON) or obese (OB) diet 60 d prior to conception through term. Diets consisted of a pelleted ration, supplemented with high quality alfalfa, calculated to provide 100% (CON) or 150% (OB) of the nutritional requirements recommended by the National Research Council [25] (NRC). Diets were maintained until necropsy or through term. After parturition, all ewes were given *ad libitum* access to high-quality alfalfa hay and were supplemented with corn to meet NRC recommendations for a lactating ewe. Weaning, in necessary groups, occurred at 4 months (PND 120) Due to incidental sample size constraints in male offspring, only female F1 were assessed in the current study. These methods were replicated to produce three separate cohorts, allowing the assessment of three developmental time points; fetal (0.9 of gestation), juvenile (2.5 months after birth), and advanced age (9 years old). Pregnant ewes were sacrificed at 0.9 gestation (d135; term 150 d) for fetal tissue collection for n = 5 CONF1 and n = 6 OBF1 (Sedation and euthanasia was performed using ketamine, isoflurane, and exsanguination). All other ewes were allowed to lamb unassisted, and F1 were housed together with their age group and maintained on 100% NRC recommendations throughout life. At 2.5 months of age blood pressure and echocardiograms were recorded for n = 9 CONF1 and n = 6 OBF1 lambs. Similarly, blood pressure and echocardiograms were recorded in n = 5 CONF1 and n = 8 OBF1 ewes at 9 years of age.

### Histochemistry

For fetal tissue assessment, fetal aortas were collected, placed in a tissue cassette (Tissue Tek; Miles Labs, Elkhart, IN), fixed with 4% paraformaldehyde in a phosphate buffer (pH 7.4; 0.12 M), and paraffin embedded as previously described [23]. Aorta sections (5 μm) were taken from the descending aorta midway through the abdomen from each fetus for staining. Sections were deparaffinized, and stained using Van Gieson, Masson Trichrome, and hematoxylin and eosin staining protocols according to manufacturer instructions (Millipore sigma, St. Louis, MO, USA). Sections were imaged using the stitched image function of CellSens software, using a 4x objective on an Olympus BX51 and a Qimage Retiga EXi camera. All histology analysis was done by a single, blinded technician. Three consecutive sections were analyzed for wall thickness measurements which were recorded using ImageJ software. Six locations equally spaced apart along each aorta section produced an average thickness estimate (6 measurements * 3 sections– 18 measurements/subject). Color deconvolution of all images was accomplished using ImageJ software after selecting the corresponding staining analysis, and mean intensity of elastin (Colour 2 image—red) and collagen (Colour 2 image -blue) expression were determined using the measure command. Optical Density (OD) was determined using $OD = \log_{10}(\text{mean intensity/max intensity})$, where max intensity = 255 (for 8 bit images).

### Blood pressure

Animals were moved from their housing pens to a smaller confinement area. Three lambs were taken at a time from the group and haltered to a stationary trimming stand to restrict movement and allow access to their ears. The dorsal side of the ear was shaved with livestock clippers, scrubbed thoroughly with iodine and 70% ethanol, dried, and treated with lidocaine gel (2%) to numb the area. An experienced technician catheterized the auricular artery with a 24 gauge x 1" catheter. The catheter was held in place by the technician while a second

technician attached a sterile extension tube, primed with saline, from the catheter to the transducer of the blood pressure monitor (BM5–VET patient monitor, Grady Medical Systems, Murrieta, CA, USA). Once the blood pressure monitor was displaying steady and consistent blood pressure readings (about 30 seconds), three separate readings, 1 minute apart, of systolic blood pressure (SBP), diastolic blood pressure (DBP), and heart rate (HR) were recorded. After successful data recording, catheters were removed and the area was wiped clean with ethanol, and light pressure was applied with gauze until bleeding stopped. Animals were returned to their housing pens, and three new animals were brought to the stand for assessment. This cycle continued until all animals were assessed. Mean arterial pressure (MAP) was determined as: MAP = [(2*DBP)+SBP]/3. Pulse pressure (PP) was determined as: PP = SBP-DBP.

## Echocardiogram

Echocardiograms were recorded on conscious animals, laying on their right side, restrained on a custom-built table. Animals were restrained using nylon ropes around their hind- and forelimbs and secured to the table using nylon netting and adjustable straps. The observation table had a small (12-inch diameter) hole in the bottom that allowed the transducer to access the animal. In vivo cardiovascular performance was recorded using a Hewlett Packard Sonos 5500 echocardiography unit (Hewlett Packard, Palo Alto, CA, USA). A 2.0–4.0 MHz cardiovascular specific probe was placed in the parasternal, short axis orientation to record the LV and RV systolic and diastolic heart dimensions. Three loops of M-mode data were recorded for each animal, and data were averaged from 3 beat cycles per loop to determine fractional shortening (FS), systolic and diastolic interventricular septum thickness (IVSs, IVSd), systolic and diastolic left ventricle internal diameter (LVIDs, LVIDd), and systolic and diastolic left ventricle posterior wall thickness (LVPWs, LVPWd). Heart rate was calculated from the time it took for three beat cycles, and used to calculate CO. End-systolic volume (ESV), end-diastolic volume (EDV), stroke volume (SV), ejection fraction (EF), and cardiac output (CO) were calculated from M-mode data using the following equations:

- $ESV = \dfrac{7}{2.4 + LVIDs} * LVIDs^3$
- $EDV = \dfrac{7}{2.4 + LVIDd} * LVIDd^3$
- $SV = EDV - ESV$
- $EF = \dfrac{SV}{EDV} * 100$
- $CO = SV * HR$

## Statistical analysis

Data were analyzed using the MIXED procedure in SAS (SAS Inst. Inc, Cary, NC, USA) with treatment in the model statement. When interactions were not significant they were removed from the final model. Differences in least square means were determined using the PDIFF statement in the model. Significant differences were determined at $p < 0.05$, and tendencies at $p < 0.1$. Results are presented as mean ± SEM.

# Results

## Histochemistry

Aortic measurements revealed that aortic walls in OBF1 fetuses were thicker ($p < 0.05$) than in CONF1 (610.5 ± 51.81 μm vs. 754.1 ± 56.52 μm). Masson trichrome staining (Fig 1A and 1B)

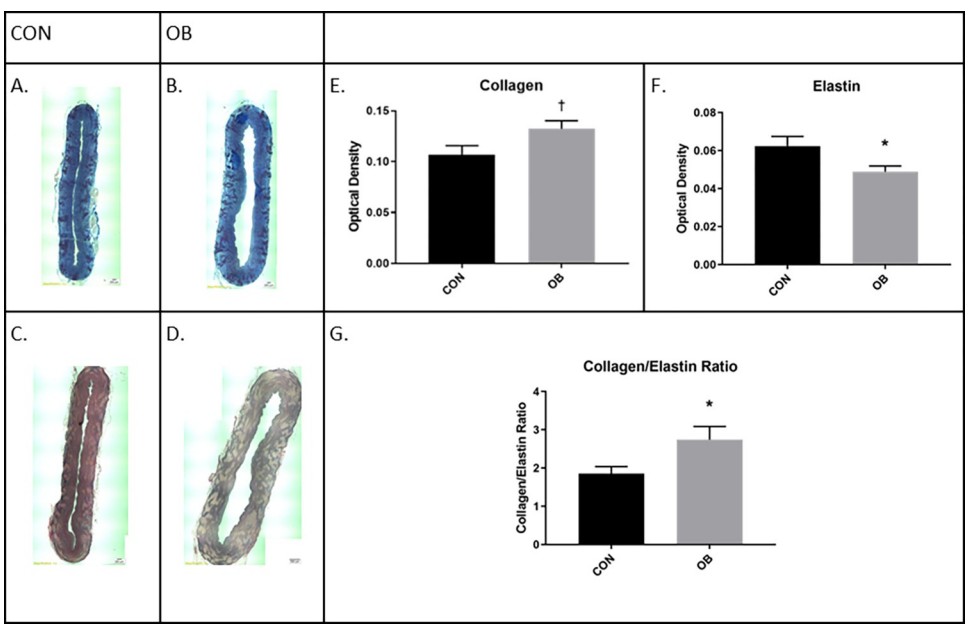

**Fig 1. Fetal aorta sections.** Masson trichrome and Van Gieson staining results from fetal aorta sections for CONF1 (n = 5) and OBF1 (n = 6) fetal aorta sections at dG 135. † means show a trend (p<0.1) between groups. *means differ (p<0.05) between groups.

revealed a tendency (p = 0.063) for greater collagen density in OBF1 aortas (0.11 ± 0.009 vs. 0.13 ± 0.008). Van Gieson staining (Fig 1C and 1D) revealed lower (p<0.05) density of elastin in OBF1 aortas relative to CONF1 (0.62 ± 0.005 vs. 0.049 ± 0.003). Together, there was 48% increase (p<0.01) in the aortic collagen:elastin ratio in OBF1 fetuses, relative to CONF1 fetuses (1.9 ± 0.18 vs. 2.7 ± 0.35, respectively).

## Blood pressure

Juvenile OBF1 exhibited elevated (p<0.05) SBP relative to CONF1 (101.2 ± 1.51 vs. 107.9 ± 3.42 mmHg for CONF1 vs OBF1, respectively), while DBP was not different between groups (81.9 ± 2.5 vs. 84.3 ±3.1 mmHg for CONF1 vs OBF1, respectively). Juvenile MAP, PP, and HR were also similar (Table 1).

Aged OBF1 ewes showed decreased (p<0.05) SBP and DBP relative to CONF1, resulting in decreased MAP (Table 2). No treatment differences were found in aged ewe PP or HR (Table 2).

**Table 1. F1 lambs invasive blood pressure.** Measurements for CONF1 (n = 9) and OBF1 (n = 6) lambs at 2.5 months of age.

|  | CONF1 | OBF1 |
|---|---|---|
| SBP | 101.2 ± 1.51[a] | 107.9 ± 3.42[b] |
| DBP | 81.96 ± 2.5 | 84.3 ± 3.1 |
| MAP | 87.8 ± 2.27 | 92.14 ± 2.68 |
| PP | 20.5 ± 2.11 | 23.63 ± 2.67 |
| HR | 127.5 ± 6.28 | 136.9 ± 7.41 |

Data are presented as mean ± standard error of the mean (SEM), and p-value from a one-tailed t-test. Systolic blood pressure (SBP). Diastolic blood pressure (DBP). Mean arterial pressure (MAP). Pulse Pressure (PP). Heart rate (HR). [a,b] means with different superscripts differ (p<0.05) within a measurement.

**Table 2. Aged F1 ewe invasive blood pressure.** Measurements for CONF1 (n = 5) and OBF1 (n = 8) ewes at 9 years of age: Treatment analysis results.

|  | CONF1 | OBF1 |
|:---:|:---:|:---:|
| SBP | 127.9 ± 4.71[a] | 112.5 ± 3.73[b] |
| DBP | 97.7 ± 4.65[a] | 86.6 ± 3.68[b] |
| MAP | 107.7 ± 4.0[a] | 97.3 ± 3.38[b] |
| PP | 30.2 ± 4.07 | 23.5 ± 3.22 |
| HR | 77.1 ± 4.53 | 66.7 ± 3.83 |

Data are presented as mean ± standard error of the mean (SEM), and p-value from a one-tailed t-test. Systolic blood pressure (SBP). Diastolic blood pressure (DBP). Mean arterial pressure (MAP). Pulse Pressure (PP). Heart rate (HR). [a,b] means with different superscripts differ (p<0.05) within a measurement.

## Echocardiography

Short axis echocardiography revealed no treatment differences in juvenile lambs (Fig 2). Further, no differences in ESV, EDV, SV or EF were found between lamb treatment groups (Fig 3). In aged ewes, no differences were found in HR or diastolic parameters, but OBF1 ewes had greater (p<0.05) LVIDs, a trend (p<0.1) for reduced IVSs, and reduced (P<0.05) LVPWs, FS, and CO (Fig 4). ESV trended (p<0.1) toward an increase in OBF1 ewes, but no differences were found in EDV between treatment groups (Fig 5). Together, ESV and EDV resulted in a trend (p<0.07) for decreased SV (80.3 ± 8.8 vs. 62.4 ± 7.5 mL), and a reduced (p<0.01) EF (83.7 ± 3.1 vs. 71.6 ± 2.9%) (Fig 5).

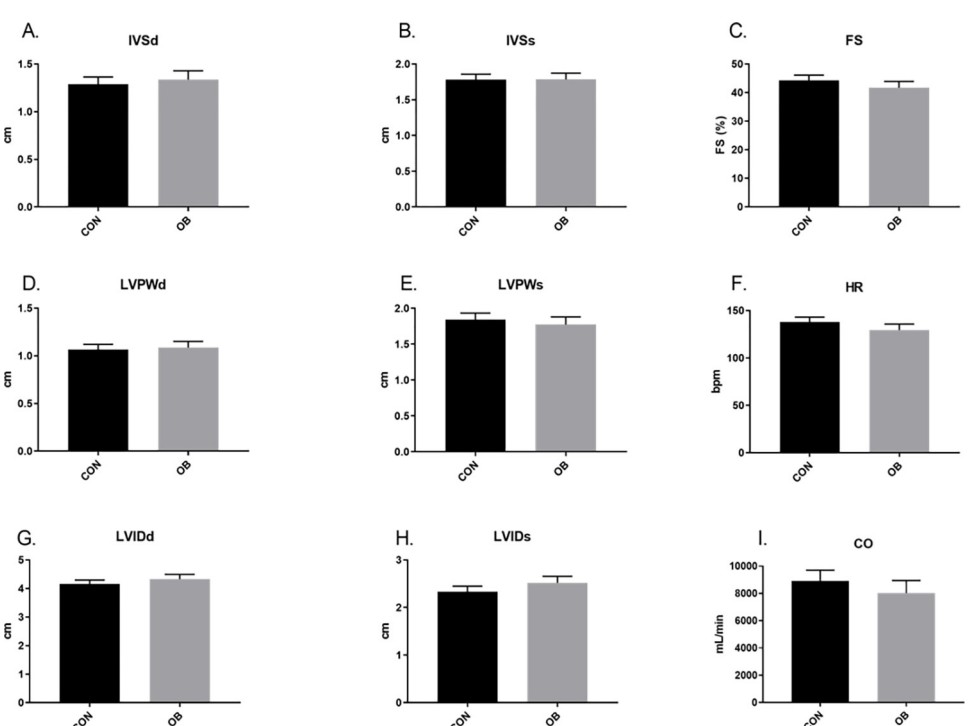

**Fig 2. Short axis echocardiography results for 2.5-month-old F1 lambs.** CONF1 (n = 9) and OBF1 (n = 6) lambs. Systolic and diastolic interventricular septum thickness (IVSs, IVSd). Systolic and diastolic left ventricle internal diameter (LVIDs, LVIDd). Systolic and diastolic left ventricle posterior wall thickness (LVPWs, LVPWd). Fractional shortening (FS). Heart rate (HR). Cardiac output (CO). † means show a trend (p<0.1). *means differ (p<0.05).

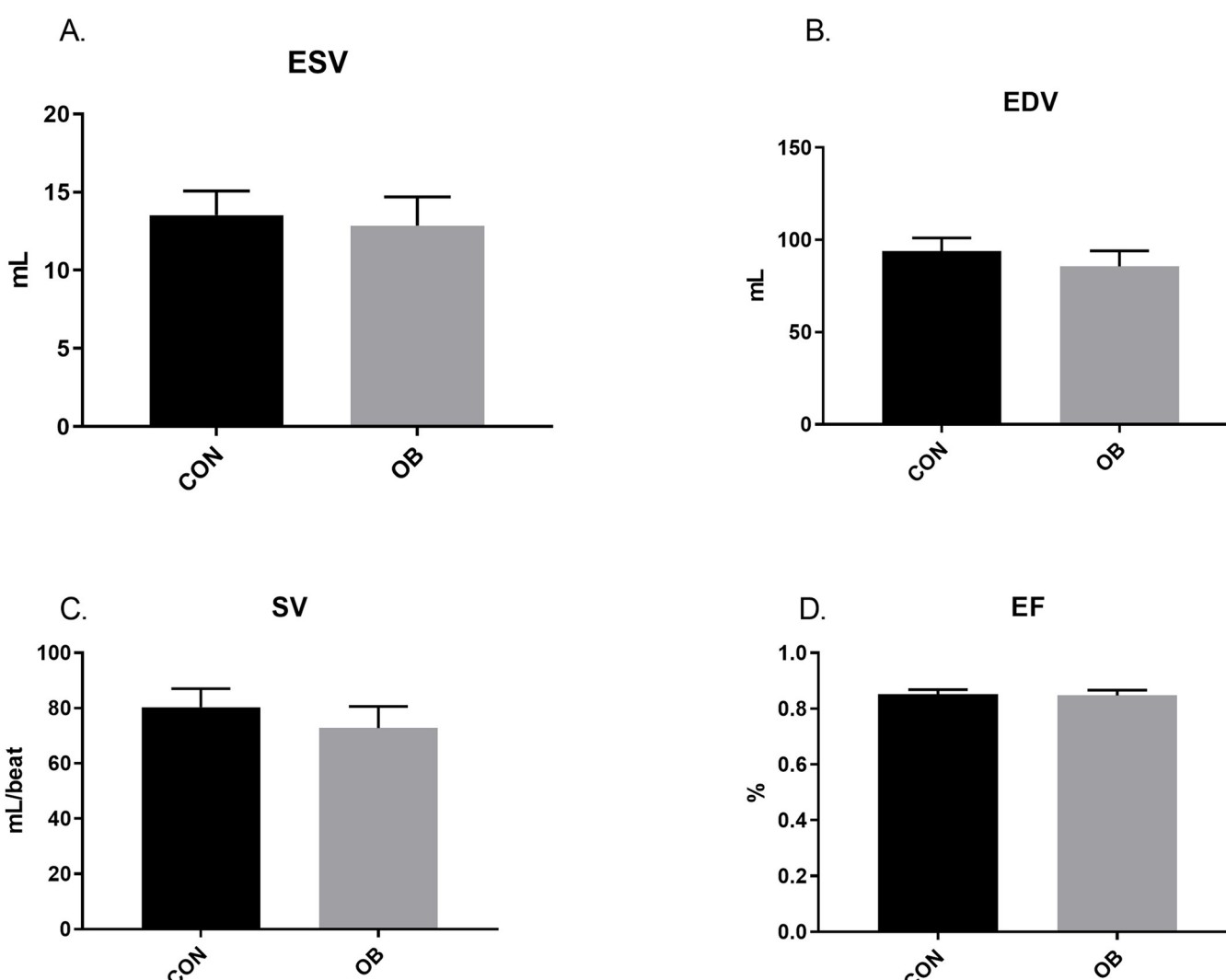

**Fig 3. Echocardiography calculations for 2.5-month-old F1 lambs.** CONF1 (n = 9) and OBF1 (n = 6) lambs. End-systolic volume (ESV). End-diastolic volume (EDV). Stroke volume (SV). Ejection fraction (EF). † means show a trend (p<0.1). *means differ (p<0.05).

## Discussion

Novel to this study, we demonstrate vascular remodeling in OBF1 fetuses, as the collagen:elastin ratio and aortic wall thickness is markedly increased in OBF1. The collagen:elastin ratio is a key determinant of aortic elasticity, increases in response to mechanical stress, and is positively correlated with cardiovascular risk factors [26, 27]. Although the mechanisms are debated between animal models [28], prior studies in sheep have demonstrated that cortisol is a potent regulator of vascular collagen and elastin synthesis [29]. Therefore, increased cortisol signaling is likely a primary mechanism for the vascular remodeling in observed in OBF1 fetuses, as we have previously demonstrated hypercortisolemia in maternal and fetal plasma throughout gestation in OB animals [30].

The myocardial hypertrophy previously observed in OBF1 fetuses may be augmented by decreased aortic compliance similar to what is seen in transverse aortic constriction models [31], as the aorta is less capable of reducing cardiovascular strain during systole. The aorta also serves as a passage for blood flow and as a reservoir that maintains blood flow during diastole.

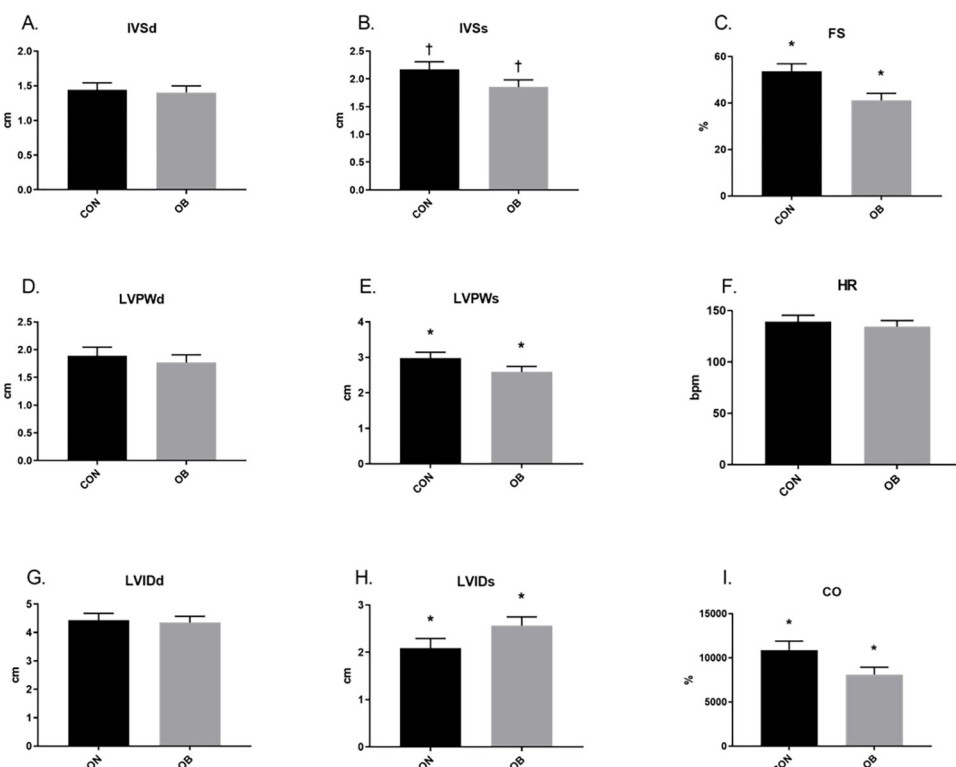

**Fig 4. Short axis echocardiography results for 9-year-old F1 ewes.** CONF1 (n = 5) and OBF1 (n = 8) ewes. Systolic and diastolic interventricular septum thickness (IVSs, IVSd). Systolic and diastolic left ventricle internal diameter (LVIDs, LVIDd). Systolic and diastolic left ventricle posterior wall thickness (LVPWs, LVPWd). Fractional shortening (FS). Heart rate (HR). Cardiac output (CO). † means show a trend (p<0.1). *means differ (p<0.05).

The "windkessel function" describes the aortas ability to absorb systolic output via elastic expansion, and maintain propulsion of blood during diastole. A recent study demonstrated that the windkessel function is impaired in baboon models of maternal diet-induced fetal programming [28]. Normally, the windkessel function reduces the stress on the left ventricle by reducing afterload. When aortic compliance is reduced, the aorta cannot expand to its full potential, leading to increased luminal pressure (as defined by Poiseuilles law: Flow Rate = (Pressure* 〚Radius〛 ^4)/(Viscosity*Length)). Increased arterial pressure, or increased afterload, has long been known to induce hypertrophic cardiomyopathy as a response to increased workload [32, 33] and as a mechanism to reduce myocardial wall stress (LaPlace's law: Intraventricular pressure = 2*(wall thickness)/radius). Data from the present study suggests that increases of the fetal aortic collagen:elastin ratio may be increasing the workload of the fetal heart, contributing to the previously observed cardiac hypertrophy in OBF1 fetal hearts, as hemodynamics are a primary determinant of fetal cardiac development [34].

Our echocardiography data suggests that the changes previously observed in fetal cardiac morphology [21–24] normalize by 2.5 months of age. This is not entirely surprising given that many other programmed phenotypes that we have observed in the fetal stages have seemingly normalized after birth [14, 24]. However, these OBF1 lambs still exhibited cardiovascular aberration, as their systolic blood pressure was greater than CONF1 lambs. These data therefore suggest that OBF1 lamb hearts may be exposed to a greater workload both pre- and postnatally. Even a mild hypertensive state over the life course can have significant effects on cardiometabolic health outcomes, as epidemiologic data suggests that increased SBP is a strong predictor of future heart disease [35].

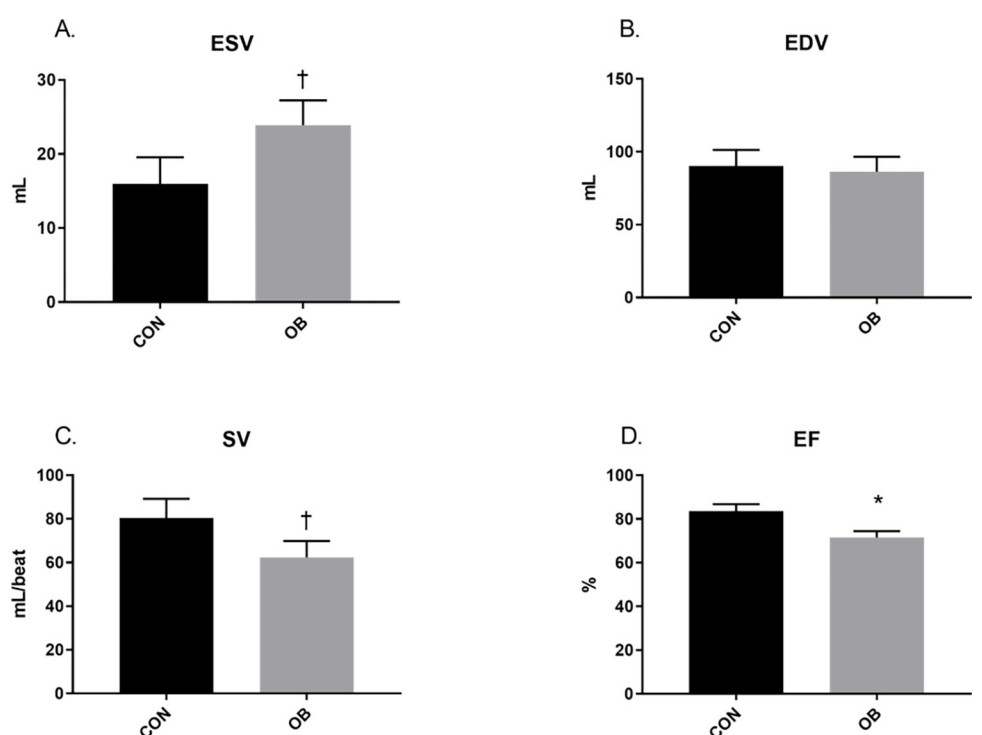

**Fig 5. Echocardiography calculation results for 9-year-old ewes.** CONF1 (n = 5) and OBF1 (n = 8) ewes. End-systolic volume (ESV). End-diastolic volume (EDV). Stroke volume (SV). Ejection fraction (EF). † means show a trend (p<0.1). *means differ (p<0.05).

Echocardiography also showed that aged OBF1 ewe hearts are morphometrically and functionally altered as IVSs, LVPWs, EF, FS, and SV were lower, and LVIDs was greater in OBF1 ewes relative to CONF1. Additionally, advance age OBF1 exhibited decreased SBP, DBP, MAP, EF, and tendencies for increased ESV and decreased SV. With these data suggesting changes in the opposite directions of what we observed at the juvenile life stage, the most plausible scenario is that the increased cardiovascular strain in the developmental and early life stages eventually progresses in to a phenotype consistent with symptoms of cardiac failure, and dilated cardiomyopathy [36]. Evidence to support this scenario can be taken from epidemiological studies, where hypertension earlier in life preludes future heart failure [37]. The early life increases in OBF1 SBP preluded decreased SBP, DBP, and MAP in advanced age OBF1 ewes, suggesting the hearts of aged OBF1 ewes may be failing to maintain normal cardiac function.

It is important to consider the strengths and limitations of the current study. Foremost, this study employs the use of a large animal model, with many similarities to humans in terms of cardiovascular and reproductive physiology. Further, it looks at multiple timepoints throughout the lifespan, allowing both for developmental and age-related changes in response to maternal obesity, and ties together investigated time points in our previous studies. However, this study is limited only to the beginning and near-ending of the life span. Most of our prior studies in this model demonstrate that offspring phenotypes from OBF0 ewes are indistinguishable from controls until they're subjected to a metabolic stress such as an ad libitum feeding trial, or pregnancy [16, 38]. The timepoints in this study were specifically chosen to examine ages at which the animals were most likely to exhibit phenotypic differences, but still lack many years of the lifespan that may provide important data to better understand how

programmed phenotypes may change over the life course. Additionally, the blood pressure and echocardiography techniques have limitations that require consideration. Although these animals were carefully handled and very used to interactions with technicians, a stress response to these procedures can still be anticipated. Therefore, the cardiovascular performance is unlikely at "baseline" as sympathetic signaling is likely altering the hemodynamics in these animals. Fortunately, animals in both groups were handled identically, allowing for group comparisons. Finally, only females were examined in this study. Given that cardiovascular outcomes differ in response to sex-specific endocrinology [39], these data may not accurately represent male physiology.

## Conclusions

This study demonstrates that in sheep, maternal obesity programs cardiovascular risk factors at the fetal stage that have consequences in later life. The increase in fetal collagen:elastin ratio shows that aortic composition is changed in OBF1 fetuses. The collagen:elastin ratio is a key factor in determining the aortas ability to reduce afterload via elastic expansion. When the collagen:elastin ratio increases, aortic distensibility is decreased. With increased aortic stiffness, the fetal heart has to work harder to eject blood, predisposing cardiac hypertrophy, as observed in prenatal OBF1 fetuses, in response to the increased workload (Wang et al., 2010). F1 lambs show similar cardiac morphology, suggesting normalization of cardiac structure at a young age. However, OBF1 lambs exhibit elevated SBP, indicating that their hearts still face increased strain after birth relative to CONF1. Upon reaching advanced age, OBF1 exhibit phenotypes consistent with heart failure. Together, these data show that cardiovascular structure and function is altered in early development resulting from maternal obesity, and these early developmental changes have consequences later in life.

## Acknowledgments

The authors would like to acknowledge students and staff of the Center for the Study of Fetal Programming for their assistance in animal care and data collection.

## Author Contributions

**Conceptualization:** Christopher L. Pankey, Qiurong Wang, Stephen P. Ford.

**Data curation:** Christopher L. Pankey, Qiurong Wang.

**Formal analysis:** Christopher L. Pankey.

**Funding acquisition:** Stephen P. Ford.

**Investigation:** Christopher L. Pankey, Qiurong Wang, Stephen P. Ford.

**Methodology:** Christopher L. Pankey, Qiurong Wang, Stephen P. Ford.

**Project administration:** Christopher L. Pankey, Stephen P. Ford.

**Resources:** Stephen P. Ford.

**Supervision:** Stephen P. Ford.

**Writing – original draft:** Christopher L. Pankey, Jessica King.

**Writing – review & editing:** Christopher L. Pankey, Jessica King.

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
