## [Decision Letter · Decision Letter 0]

4 May 2022

PONE-D-22-06151Cardiovascular consequences of maternal obesity throughout the lifespan in first generation sheepPLOS ONE

Dear Dr. Pankey,

Thank you for submitting your manuscript to PLOS ONE. After careful consideration, we feel that it has merit but does not fully meet PLOS ONE’s publication criteria as it currently stands. Therefore, we invite you to submit a revised version of the manuscript that addresses the points raised by the reviewers during the review process.

We look forward to receiving your revised manuscript.

Kind regards,

Christopher Torrens

Academic Editor

PLOS ONE

Journal Requirements:

“This study was supported by the Dual Purpose with Dual Benefits grant from the National Institutes of Health (https://grants.nih.gov/funding/index.htm), grant number R01 HD070096-01A1, awarded to SPF.”

Reviewers' comments:

Reviewer's Responses to Questions

**Comments to the Author**

1. Is the manuscript technically sound, and do the data support the conclusions?

Reviewer #1: Yes

Reviewer #2: Yes

2. Has the statistical analysis been performed appropriately and rigorously? 

Reviewer #1: Yes

Reviewer #2: I Don't Know

3. Have the authors made all data underlying the findings in their manuscript fully available?

Reviewer #1: Yes

Reviewer #2: Yes

4. Is the manuscript presented in an intelligible fashion and written in standard English?

Reviewer #1: Yes

Reviewer #2: Yes

5. Review Comments to the Author

Reviewer #1: The study by Pankey et al examine the short and long-term cardiovascular effects of maternal obesity in a sheep model. Ewes were fed an obesogenic diet prior to and during pregnancy and offspring were examined in late gestation, young life and old age. Offspring from obese mothers had an increased aortic collagen:elastin ratio in fetal life, systolic blood pressure was increased in young life but decreased in old age. Echocardiography revealed changes in old age, notably a reduced ejection fraction. The authors suggest the increase in cardiac workload in early life led to features of heart failure in later life.

The study complements other work performed with this maternal obesity model and the authors should be commended for studying offspring to such an old age.

Comments:

- Line 66 The introduction would benefit from briefly stating some more details of the well-characterised model, for context, including more specifics of the cardiovascular outcomes previously observed in fetal and mid-life. In part the rationale of the study was to examine effects at different ages not previously reported (the Discussion starts with this instead).

- Line 56 onwards –the use of ‘OB’ to refer to the offspring of different ages and using ‘ewe’ for both mothers and offspring is slightly confusing. ‘OB ewes’ reads to me as the mothers so it is harder to distinguish maternal and offspring outcomes.

- There is a lack of detail about the experimental protocol. Although this is a well utilised model, the manuscript would benefit from some more basic information eg. relative obesity achieved by the OB diet in pregnancy, body weights/composition at each age.

- Blood pressure measurements.

• The wording here (line 110) suggests measurements were taken on 3 animals when it is clear that 6-9 animals were recorded. Please clarify what was done.

• Line 117 Was there any validation done about stablisation of blood pressure over a longer period given the invasive nature of the blood pressure measurement? 30 seconds seems very short. Could the authors comment on the potential effect of stress of the blood pressure and echocardiogram procedures on the results?

• It is not clear whether the same animals were used at 2.5 months and 9 years. It should state that the catheter was removed after the measurement at 2.5 months.

- It is not clear whether the same animals were used at 2.5 months and 9 years. It should state that the catheter was removed after the measurement at 2.5 months.

- The authors propose that “increased cardiovascular strain in the developmental and early life stages eventually progresses in to a phenotype consistent with symptoms of cardiac failure”. It would have been interesting to have aortic measurements from the 9 year-old animals. Could the authors provide any more evidence that the changes seen in fetal life persisting till old age in this model or similar?

- The authors should comment on the potential role of sex, given that only female offspring were studied, and in reference to the previous work with this model. There was no information about the sex ratio in the fetal cohort.

Minor comments:

- Line 145 states sex was included in the model but only female sheep were examined

- Line 163 should read ‘SBP relative to CONF1’ not F2

- I suggest using the term juvenile for the lambs at 2.5 months rather than neonatal. This age is no longer neonatal.

Reviewer #2: The study from Pankey and colleagues hypothesized that diet-induced obesity throughout pregnancy would cause cardiovascular dysfunction in neonatal and adult offspring.

In the first paragraph of the introduction it is not clear whether this is referring to offspring of an obese mother or of individuals whom are obese. Although the next paragraph is clearly about offspring it would be pertinent to make it quite clear that the first is about an individual risk of cardiovascular disease. Further in the 4th paragraph it is not explicitly obvious whether the ewes that are referenced are pregnant during the experiments in which the data is reported.

Line 76 I would rearrange the words fetuses and lambs. The way it is written is confusing it is possible that one might interpret this as the lamb getting pregnant to the produce a fetus (F2).

Could you elaborate on the sample size constraints regarding male offspring? Was this as a result of in utero death in the OB group?

It is my opinion that ‘neonatal’ is not an appropriate name for the 2.5 month group, this age is significantly greater than what could be deemed to be neonatal.

At what age were the lambs weaned? Was ewe milk collected for assessment of caloric intake?

How were the samples sizes determined? N=5 seems low for ovine studies.

I believe it is unnecessary to refer to the lambs as F1 – there is no F2 and therefore it merely creates unnecessary confusing.

Precisely where were the aorta sections taken?

Line 94 and 115. This section should not be referred to as immunohistochemistry. There are no immunohistochemical methods described in this section.

Were the lambs frequently handled? I am curious because I worry the lambs would have been highly stressed during the blood pressure and echocardiography recordings and that this may have impacted the blood pressure recordings.

The biometry of the cohorts should be described, particularly as the weight is referred to in the discussion.

The ethics are described in the submission (and apologies if I missed it) but not in the manuscript itself.

Is Line 153-6 referring to fetuses? Could you also please report on the 2.5 month and 9 year data?

Fig 1. Annotation of significance is only required on one group.

Was there any difference in trajectory of cardiovascular outcomes over time between groups?

6. PLOS authors have the option to publish the peer review history of their article (what does this mean?). If published, this will include your full peer review and any attached files.

Reviewer #1: No

Reviewer #2: **Yes: **Beth Allison

---

## [Author Response · Author response to Decision Letter 0]

20 Jun 2022

The text below is also included as an uploaded file, and is color coded in that version for ease of reading. 

PONE-D-22-06151

Cardiovascular consequences of maternal obesity throughout the lifespan in first generation sheep

PLOS ONE

Dear Dr. Pankey,

Thank you for submitting your manuscript to PLOS ONE. After careful consideration, we feel that it has merit but does not fully meet PLOS ONE’s publication criteria as it currently stands. Therefore, we invite you to submit a revised version of the manuscript that addresses the points raised by the reviewers during the review process.

We look forward to receiving your revised manuscript. 

Kind regards,

Christopher Torrens

Academic Editor

PLOS ONE

Journal Requirements: 

“This study was supported by the Dual Purpose with Dual Benefits grant from the National Institutes of Health (https://grants.nih.gov/funding/index.htm), grant number R01 HD070096-01A1, awarded to SPF.”

Reviewers' comments:

Reviewer's Responses to Questions 

Comments to the Author

1. Is the manuscript technically sound, and do the data support the conclusions?

Reviewer #1: Yes

Reviewer #2: Yes

2. Has the statistical analysis been performed appropriately and rigorously? 

Reviewer #1: Yes

Reviewer #2: I Don't Know

3. Have the authors made all data underlying the findings in their manuscript fully available?

Reviewer #1: Yes

Reviewer #2: Yes

4. Is the manuscript presented in an intelligible fashion and written in standard English?

Reviewer #1: Yes

Reviewer #2: Yes

5. Review Comments to the Author

Dear reviewers: Please note that the updated line numbers referenced in our responses are in the “Revised Manuscript with Track Changes”, and the revisions must be showing for the line numbers to be accurate. 

Reviewer #1: The study by Pankey et al examine the short and long-term cardiovascular effects of maternal obesity in a sheep model. Ewes were fed an obesogenic diet prior to and during pregnancy and offspring were examined in late gestation, young life and old age. Offspring from obese mothers had an increased aortic collagen:elastin ratio in fetal life, systolic blood pressure was increased in young life but decreased in old age. Echocardiography revealed changes in old age, notably a reduced ejection fraction. The authors suggest the increase in cardiac workload in early life led to features of heart failure in later life.

The study complements other work performed with this maternal obesity model and the authors should be commended for studying offspring to such an old age.

Comments:

- Line 66 The introduction would benefit from briefly stating some more details of the well-characterised model, for context, including more specifics of the cardiovascular outcomes previously observed in fetal and mid-life. In part the rationale of the study was to examine effects at different ages not previously reported (the Discussion starts with this instead).

Thank you for this comment. We agree, and have added lines 55-60 to address this concern. We had also discussed prior findings on lines 54-61 (now lines 63-71), but recognize how the order and wording may have misled the reader to assume that was not from our model. As such, we also reordered and revised this section of the introduction.

- Line 56 onwards –the use of ‘OB’ to refer to the offspring of different ages and using ‘ewe’ for both mothers and offspring is slightly confusing. ‘OB ewes’ reads to me as the mothers so it is harder to distinguish maternal and offspring outcomes.

Thank you for this comment, we agree and have revised all “OB” throughout the manuscript to either OBF0 or OBF1 to specifically label which generation is being discussed. Similar revisions were also done for CON subjects. 

- There is a lack of detail about the experimental protocol. Although this is a well utilised model, the manuscript would benefit from some more basic information eg. relative obesity achieved by the OB diet in pregnancy, body weights/composition at each age.

Thank you for addressing this point. In response, we have added comments throughout the manuscript and added a citation (Long et al., 2010 – citation #15) that addresses the phenotypes questioned in this remark. We feel it is best to provide citations in this case, opposed to discussing in depth, as the mentioned citation is a complete manuscript addressing your concerns and we fear it would be too much detail for the current manuscript. We also discuss the cardiovascular specific phenotypes in this model on lines 78-81 to further address this concern. 

- Blood pressure measurements.

• The wording here (line 110) suggests measurements were taken on 3 animals when it is clear that 6-9 animals were recorded. Please clarify what was done.

Thank you, additional information has been added to lines 125-126 and 134-138 to clarify this procedure. 

• Line 117 Was there any validation done about stablisation of blood pressure over a longer period given the invasive nature of the blood pressure measurement? 30 seconds seems very short. Could the authors comment on the potential effect of stress of the blood pressure and echocardiogram procedures on the results?

There were no validation procedures performed, as the stabilization period was intended to ensure the equipment was working properly, opposed to waiting for the animals to reach “baseline”. Indeed, we agree this needs addressed and have added comments to the limitations in the discussion (lines 291-296).

• It is not clear whether the same animals were used at 2.5 months and 9 years. It should state that the catheter was removed after the measurement at 2.5 months.

Different animals were used for each cohort, as we outline on lines 98-100: “These methods were replicated to produce three separate cohorts, allowing the assessment of three developmental time points; fetal (0.9 of gestation), juvenile (2.5 months after birth), and advanced age (9 years old)”. 

We have added details for the blood pressure procedure on lines 134-138 to address the second concern. 

- The authors propose that “increased cardiovascular strain in the developmental and early life stages eventually progresses in to a phenotype consistent with symptoms of cardiac failure”. It would have been interesting to have aortic measurements from the 9 year-old animals. Could the authors provide any more evidence that the changes seen in fetal life persisting till old age in this model or similar?

We cannot currently provide more information in this model, but efforts are under-way to continue examining the 9-year-old ewes, and tissue samples will eventually be collected at death/sacrifice to specifically address this query. Hopefully, these efforts will generate additional data to increase our understanding of this progression. Since we cannot currently provide these data, we have cited evidence to support our rationale (citation 37 – line 276) 

- The authors should comment on the potential role of sex, given that only female offspring were studied, and in reference to the previous work with this model. There was no information about the sex ratio in the fetal cohort.

Thank you for providing this critique. In response, we have added a comment about the potential role of sex-specific endocrinology that may influence cardiovascular health, and provided citation #40 to support this point (lines 296-298). 

Minor comments:

- Line 145 states sex was included in the model but only female sheep were examined

Thank you for catching this error. We have revised the statement appropriately (now line 163) to remove the term.

- Line 163 should read ‘SBP relative to CONF1’ not F2

Thank you, this error has been revised (now line 181)

- I suggest using the term juvenile for the lambs at 2.5 months rather than neonatal. This age is no longer neonatal.

Thank you for this suggestion. We concur that juvenile is more accurate, and have replaced that term throughout the manuscript. 

Reviewer #2: The study from Pankey and colleagues hypothesized that diet-induced obesity throughout pregnancy would cause cardiovascular dysfunction in neonatal and adult offspring.

In the first paragraph of the introduction it is not clear whether this is referring to offspring of an obese mother or of individuals whom are obese. Although the next paragraph is clearly about offspring it would be pertinent to make it quite clear that the first is about an individual risk of cardiovascular disease. Further in the 4th paragraph it is not explicitly obvious whether the ewes that are referenced are pregnant during the experiments in which the data is reported.

Thank you for taking the time to review our manuscript. We have added the term “individual” in to the opening paragraph to alleviate this concern (line 34). 

Line 76 I would rearrange the words fetuses and lambs. The way it is written is confusing it is possible that one might interpret this as the lamb getting pregnant to the produce a fetus (F2).

Thank you, we have rearranged the terms in response to this comment (now line 91). 

Could you elaborate on the sample size constraints regarding male offspring? Was this as a result of in utero death in the OB group?

Thank you for this question. The sample size constraints were purely incidental, and there were no deaths or illnesses in males that contributed to these constraints. To add clarity to the text, the term incidental was included on line 97. 

It is my opinion that ‘neonatal’ is not an appropriate name for the 2.5 month group, this age is significantly greater than what could be deemed to be neonatal.

Thank you for this comment. Reviewer one had a similar concern, and we agree with you both. In response, we have substituted the term “juvenile” throughout the text to replace “neonatal”. 

At what age were the lambs weaned? Was ewe milk collected for assessment of caloric intake?

Thank you for this question. The juvenile lambs were not weaned, as weaning would typically occur at 4 months (PND 120). Milk samples were not assessed, but after parturition all ewes (both groups) were maintained on 100% NRC requirements for a lactating ewe (discussed on lines 87-89). So the experimental diet only occurred from 60 d prior to conception through term (line 84). We have added detail to line 97 to address this concern. 

How were the samples sizes determined? N=5 seems low for ovine studies.

Our power analysis on several end-points in our published studies show that we have better than 80% power to detect biologically important differences with as small as 5 animals per group (80% power, p=0.05) to discern treatment differences. A sample size of 4-6 is not uncommon in our model for publication 

(10.1152/ajpregu.00498.2009 http://dx.doi.org/10.1016/j.domaniend.2017.04.002 10.1152/ajpregu.00072.2009 10.1093/jas/sky215). 

I believe it is unnecessary to refer to the lambs as F1 – there is no F2 and therefore it merely creates unnecessary confusing.

Thank you for this comment. Although we are not studying F2, the maternal ewes are F0, and we feel this shorthand differentiates these two generations to help the reader know we are discussing the offspring and not the F0 ewes. We feel this is important in this paper as the aged F1 females could also be correctly termed “ewes”. These classifications are also consistent with our prior publications within the same model, which is convenient if reading multiple studies from our group. 

Precisely where were the aorta sections taken?

Thank you for this important question. We have revised lines 111-112 to add this detail.

Line 94 and 115. This section should not be referred to as immunohistochemistry. There are no immunohistochemical methods described in this section.

Thank you for noting this error. We have changed the term to the more appropriate “histochemistry” (now lines 108 and 169). 

Were the lambs frequently handled? I am curious because I worry the lambs would have been highly stressed during the blood pressure and echocardiography recordings and that this may have impacted the blood pressure recordings.

Thank you for this important question. Reviewer one had similar concerns, and we have added text to lines 291-296 to address these concerns. 

The biometry of the cohorts should be described, particularly as the weight is referred to in the discussion.

In response to comments from reviewer one, the text we believe you may be referencing has been moved to the introduction (lines 79-81). These findings were in prior studies, and therefore reporting the data and statistics here would not be conventional, so we have only mentioned the broad outcomes and provided the citations. 

We recognize that we may be misinterpreting your intended critique here also, and apologize if that is the case. We are certainly willing to revisit this discussion if we are not adequately addressing your concern. 

The ethics are described in the submission (and apologies if I missed it) but not in the manuscript itself.

Thank you for this note, we were not certain by reading the author directions if it was supposed to be in both, so we removed details from the original submission. We now have these descriptions in the manuscript on lines 89-90, and 102-103.

Is Line 153-6 referring to fetuses? Could you also please report on the 2.5 month and 9 year data?

Yes, those results are referring to the fetal tissues that were collected at necropsy (now line 170: “OBF1 fetuses were…”). We do not have tissue samples yet from the juvenile or aged ewe cohort, as they were kept alive for further assessment (lines 102-106 describe these cohorts).

Fig 1. Annotation of significance is only required on one group.

Thank you, we have edited Fig. 1 accordingly. 

Was there any difference in trajectory of cardiovascular outcomes over time between groups?

Thank you for this question. Indeed there were differences between OB and CON groups as discussed. As far as differences between OB fetuses, OB juveniles, and OB ewes, we do not yet have adequate data to compare many of the outcomes over time. Future studies are currently in preparation to address this question. 

6. PLOS authors have the option to publish the peer review history of their article (what does this mean?). If published, this will include your full peer review and any attached files.

Do you want your identity to be public for this peer review? For information about this choice, including consent withdrawal, please see our Privacy Policy.

Reviewer #1: No

Reviewer #2: Yes: Beth Allison

---

## [Decision Letter · Decision Letter 1]

24 Aug 2022

Cardiovascular consequences of maternal obesity throughout the lifespan in first generation sheep

PONE-D-22-06151R1

Dear Dr. Pankey,

We’re pleased to inform you that your manuscript has been judged scientifically suitable for publication and will be formally accepted for publication once it meets all outstanding technical requirements.

Kind regards,

Christopher Torrens

Academic Editor

PLOS ONE

Additional Editor Comments (optional):

Reviewers' comments:

Reviewer's Responses to Questions

**Comments to the Author**

1. If the authors have adequately addressed your comments raised in a previous round of review and you feel that this manuscript is now acceptable for publication, you may indicate that here to bypass the “Comments to the Author” section, enter your conflict of interest statement in the “Confidential to Editor” section, and submit your "Accept" recommendation.

Reviewer #1: All comments have been addressed

Reviewer #2: All comments have been addressed

2. Is the manuscript technically sound, and do the data support the conclusions?

Reviewer #1: Yes

Reviewer #2: Yes

3. Has the statistical analysis been performed appropriately and rigorously? 

Reviewer #1: Yes

Reviewer #2: Yes

4. Have the authors made all data underlying the findings in their manuscript fully available?

Reviewer #1: Yes

Reviewer #2: Yes

5. Is the manuscript presented in an intelligible fashion and written in standard English?

Reviewer #1: Yes

Reviewer #2: Yes

6. Review Comments to the Author

Reviewer #1: (No Response)

Reviewer #2: All my concerns have been addressed. The manuscript is suitable for publication.

I have no concerns about dual publication, research ethics or publication ethics.

7. PLOS authors have the option to publish the peer review history of their article (what does this mean?). If published, this will include your full peer review and any attached files.

Reviewer #1: No

Reviewer #2: **Yes: **Beth Allison

---

## [Editor Report · Acceptance letter]

25 Aug 2022

PONE-D-22-06151R1 

Cardiovascular consequences of maternal obesity throughout the lifespan in first generation sheep 

Dear Dr. Pankey:

I'm pleased to inform you that your manuscript has been deemed suitable for publication in PLOS ONE. Congratulations! Your manuscript is now with our production department. 

Kind regards, 

on behalf of

Dr. Christopher Torrens 

Academic Editor

PLOS ONE